# Specific modulation of CRISPR transcriptional activators through RNA-sensing guide RNAs in mammalian cells and zebrafish embryos

Oana Pelea[1]*[†], Sarah Mayes[1], Quentin RV Ferry[1][‡], Tudor A Fulga[1][§], Tatjana Sauka-Spengler[1,2]*

[1]University of Oxford, MRC Weatherall Institute of Molecular Medicine, Radcliffe Department of Medicine, Oxford, United Kingdom; [2]Stowers Institute for Medical Research, Kansas City, United States

**\*For correspondence:**
o.pelea@bioc.uzh.ch (OP);
tatjana.sauka-spengler@imm.ox.ac.uk (TS-S)

**Present address:** [†]Department of Biochemistry, University of Zurich, Winterthurerstr, Zurich, Switzerland; [‡]Massachusetts Institute of Technology, Picower Institute for Learning and Memory, Cambridge, United States; [§]Vertex Pharmaceuticals, Boston, United States

**Competing interest:** The authors declare that no competing interests exist.

## eLife assessment

The authors aim to develop a CRISPR system that can be activated upon sensing an RNA. As an initial step to this goal, they describe RNA-sensing guide RNAs for controlled activation of CRISPR modification. Many of the data look **convincing** and while several steps remain to achieve the stated goal in an in vivo setting and for robust activation by endogenous RNAs, the current work will be **important** for many in the field.

**Abstract** Cellular transcripts encode important information regarding cell identity and disease status. The activation of CRISPR in response to RNA biomarkers holds the potential for controlling CRISPR activity with spatiotemporal precision. This would enable the restriction of CRISPR activity to specific cell types expressing RNA biomarkers of interest while preventing unwanted activity in other cells. Here, we present a simple and specific platform for modulating CRISPR activity in response to RNA detection through engineering *Streptococcus pyogenes* Cas9 single-guide RNAs (sgRNAs). sgRNAs are engineered to fold into complex secondary structures that, in the ground state, inhibit their activity. Engineered sgRNAs become activated upon recognising complementary RNAs, thus enabling Cas9 to perform its function. Our approach enables CRISPR activation in response to RNA detection in both HEK293T cells and zebrafish embryos. Iterative design optimisations allowed the development of computational tools for generating sgRNAs capable of detecting RNA sequences of choice. Mechanistic investigations reveal that engineered sgRNAs are cleaved during RNA detection, and we identify key positions that benefit from chemical modifications to improve the stability of engineered sgRNAs in vivo. Our sensors open up novel opportunities for developing new research and therapeutic applications using CRISPR activation in response to endogenous RNA biomarkers.

## Introduction

Traditional methods for detecting RNA in live cells include hybridisation probes, fluorescent aptamers, and fluorescent RNA-binding proteins. Such methods enable the visualisation of RNA foci (*Mannack et al., 2016*), but they cannot drive cellular reprogramming in response to RNA detection. Recent progress has been made towards enabling the activation of therapeutically relevant payloads in response to RNA detection (*Jiang et al., 2023*). Nevertheless, linking RNA detection with gene

editing or modulation of gene expression remains a challenge. Since cellular RNAs offer crucial information about cell identity, differentiation, disease status, and environmental exposure (*Abdolhosseini et al., 2019*; *Kotliar et al., 2019*), modulating CRISPR activation in response to RNA detection holds tremendous promise for future innovations.

The single-guide RNA (sgRNA) component of *Streptococcus pyogenes* CRISPR-Cas9 systems (*Jinek et al., 2012*) tolerates extensive modifications and sgRNA engineering has been established as a method for controlling CRISPR activity. This is achieved by designing complex sgRNA secondary structures that can inactivate sgRNA function. Inactivated sgRNA structures serve as a starting point for the development of technologies that aim to control CRISPR in response to different molecular triggers (*Pelea et al., 2022*). Inactivated sgRNAs can be successfully re-activated in response to small molecules (*Tang et al., 2017*), proteins (*Ferry et al., 2017*), DNA antisense oligonucleotides (*Ferry et al., 2017*), as well as RNA (*Hanewich-Hollatz et al., 2019*; *Hochrein et al., 2021*; *Jakimo et al., 2018*; *Jiao et al., 2021*; *Jin et al., 2019*; *Li et al., 2019*; *Liu et al., 2022*; *Lin et al., 2020*; *Siu and Chen, 2019*; *Galizi et al., 2020*; *Hunt and Chen, 2022a*; *Hunt and Chen, 2022b*; *Ying et al., 2020*; *Choi et al., 2023*).

Due to the complexity of eukaryotic systems and cellular compartmentalisation, modulating CRISPR activity in response to RNA detection remains a tantalising challenge (*Hunt and Chen, 2022a*; *Hunt and Chen, 2022b*). While limited evidence is available for the modulation of CRISPR activity following RNA detection in mammalian cells (*Hanewich-Hollatz et al., 2019*; *Hochrein et al., 2021*; *Hunt and Chen, 2022a*; *Hunt and Chen, 2022b*; *Lin et al., 2020*; *Ying et al., 2020*), published technologies that rely on sgRNA engineering still require improvement to enhance the dynamic range of activation. There is no clear evidence that existing engineering approaches can be generalised to detect a diverse panel of RNA sequences. Additionally, functional validation of engineered sgRNAs in vivo remains unexplored. Furthermore, the lack of computational tools to design engineered sgRNAs for RNA detection applications and limited knowledge of the molecular mechanisms underlying RNA detection by engineered sgRNAs further compound this challenge.

Our work presents a highly specific system for modulating CRISPR transcriptional activators in response to RNA detection. Here, we show that engineered RNA-sensing iSBH-sgRNAs (inducible spacer-blocking hairpin sgRNAs, *Ferry et al., 2017*) enable modulation of CRISPR activity in eukaryotic cells as well as in developing zebrafish embryos. Native sgRNAs have two components: the spacer and scaffold sequences. The spacer sequence is complementary to the CRISPR-targeting sequence (CTS) and determines sgRNA specificity, while the scaffold sequence stabilises interactions between the sgRNA and Cas9 proteins (*Figure 1A*, *Jinek et al., 2012*; *Gaj et al., 2013*). iSBH-sgRNA designs are engineered sgRNAs that cannot drive CRISPR activity in their ground state due to their complex secondary structures. They differ from native sgRNAs by having a 14-nucleotide loop and a partially complementary spacer* sequence in addition to the spacer and scaffold sequences. The complementarity between the spacer and spacer* sequences creates a complex secondary structure that inactivates the sgRNA function (*Figure 1A*). When iSBH-sgRNAs are introduced into cells that do not express complementary RNA sequences, spacer sequences are blocked, and CRISPR activity is turned OFF. However, when RNA sequences complementary to the loop and spacer* sequences are present, the iSBH-sgRNA conformation changes, exposing spacer sequences and turning ON CRISPR activity.

After we demonstrated that iSBH-sgRNAs can activate CRISPR in response to RNA detection, we implemented a standard design-build-test Synthetic Biology cycle to select iSBH-sgRNA designs with superior performance and dynamic ranges of activation. Additionally, we introduced the MODesign algorithm, which allows users to create custom-engineered RNA-sensing iSBH-sgRNAs for their CRISPR applications. We also investigated the mechanism of iSBH-sgRNA activation and showed that RNA detection occurs through a double-stranded RNA cleavage mechanism. By studying this mechanism further, we identified key residue positions that are prone to cleavage by cellular factors and used chemical modifications to protect and stabilise engineered iSBH-sgRNAs in developing zebrafish embryos. We anticipate that our ability to control iSBH-sgRNA activity in response to RNA triggers in vivo will enable the development of novel therapeutic applications that harness endogenous RNA biomarkers to control CRISPR activity.

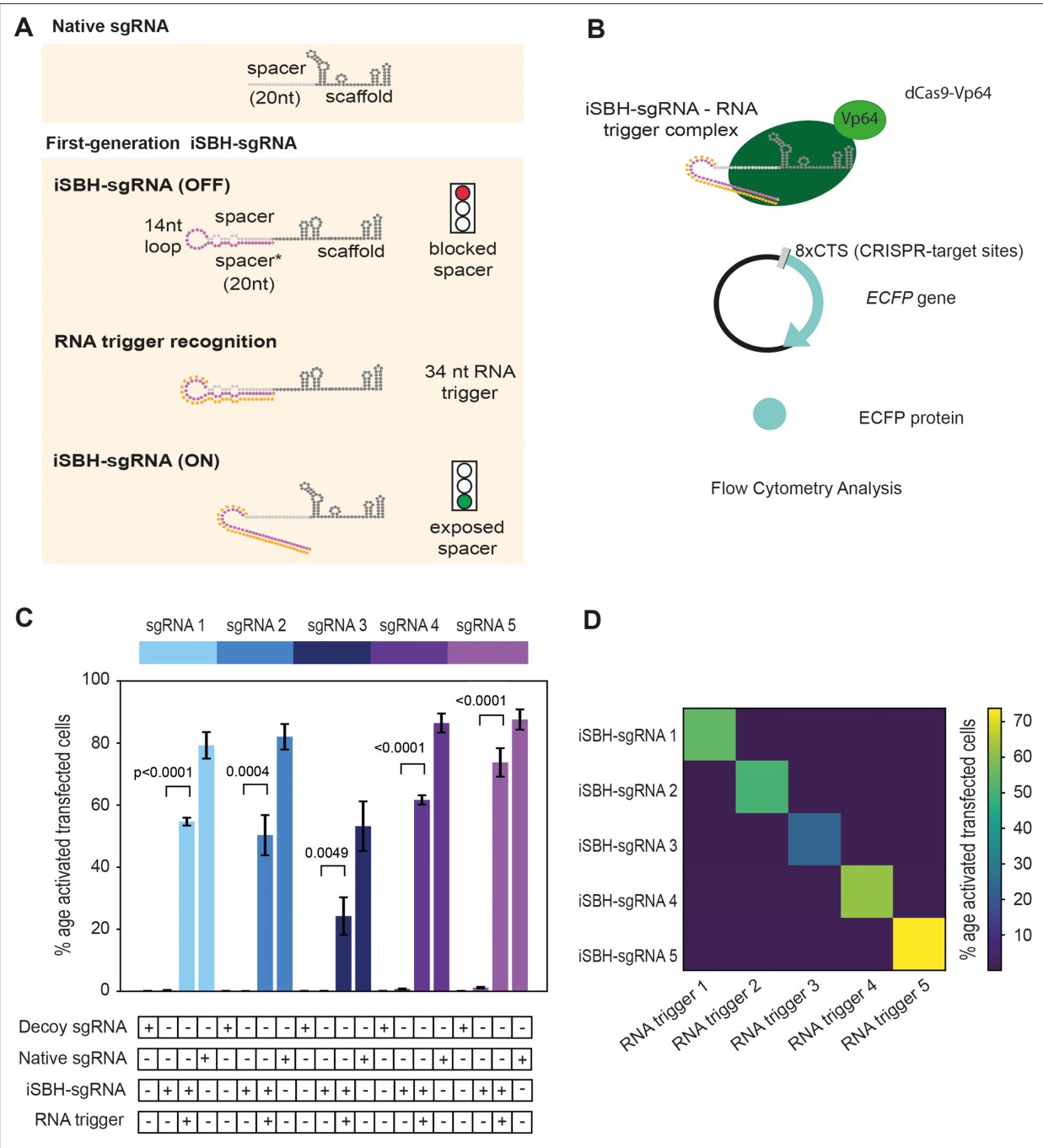

**Figure 1.** First-generation iSBH-sgRNAs detect short RNA triggers in HEK293T cells. (**A**) Native sgRNA sequences are composed of spacer and scaffold sequences (*Jinek et al., 2012*). iSBH-sgRNAs fold into complex secondary structures that interfere with the Cas9 ability to recognise target DNA sequences (OFF-state, *Ferry et al., 2017*). iSBH-sgRNAs were designed by extending the 5' end of the spacer sequence with a 14 nt loop and a spacer* sequence partially complementary with the spacer. Bulges were also introduced within the iSBH-sgRNA sequence in order to ensure that the interaction between the spacer* and RNA trigger is more energetically favourable. In the ON-state, iSBH-sgRNAs recognise complementary RNA triggers and become activated, enabling Cas9 to perform its function. Short RNA triggers are complementary with the iSBH-sgRNA loop and spacer* sequence. (**B**) Inside cells, RNA triggers are expected to bind to complementary iSBH-sgRNAs, inducing iSBH-sgRNA activation. Activated iSBH-sgRNAs are recognised by CRISPRa effectors and drive ECFP production from a fluorescent reporter. In this particular example, activated iSBH-sgRNAs interact with dCas9-Vp64 (*Maeder et al., 2013*) and drive ECFP production from an 8xCTS-ECFP reporter (*Nissim et al., 2014*). Following reporter induction, ECFP production could be monitored by Flow Cytometry. (**C**) Starting from five different sgRNA spacer sequences, we designed five different iSBH-sgRNA sequences. For each iSBH-sgRNA, corresponding RNA triggers and 8xCTS-ECFP reporters were also designed. Ability of first-generation iSBH-sgRNA designs to drive expression of the ECFP reporter was assessed in the absence or presence of complementary RNA triggers. Experiments were

*Figure 1 continued on next page*

Figure 1 continued

carried out using dCas9-Vp64 and 8xCTS-ECFP reporters. (**D**) An orthogonality test was performed, in which the five iSBH-sgRNA designs were tested against all five RNA triggers. Activation is only detected in the presence of matching iSBH-sgRNA and RNA trigger pairs. Figure shows mean ± standard deviation values measured for three biological replicates. Values above bars represent fold turn-on values for iSBH-sgRNA activation (blue) and p-values (black) determined through unpaired *t*-tests.

The online version of this article includes the following figure supplement(s) for figure 1:

**Figure supplement 1.** First-generation iSBH-sgRNAs detect short RNA triggers in HEK293T cells.

## Results

### iSBH-sgRNAs enable conditional CRISPR activation in response to RNA detection

The first aim of this study was to test whether a first generation of iSBH-sgRNA designs enabled modulation of CRISPR transcriptional activators in response to RNA detection in mammalian cells. To address this, five iSBH-sgRNAs were designed, each featuring a distinct sgRNA spacer sequence. For each iSBH-sgRNA, RNA triggers complementary to the loop, and the spacer* sequences were also designed (*Figure 1A*). To prevent their degradation by cellular nucleases, RNA triggers were flanked by 5' and 3' hairpins (a detailed description of hairpins can be found in the *Supplementary file 1*).

Mammalian plasmids containing iSBH-sgRNA and corresponding RNA triggers were constructed under the control of U6 promoters (*Paul et al., 2002*). The plasmids were co-transfected into HEK293T cells along with a CRISPR activator (CRISPRa) and a fluorescent reporter cassette to monitor CRISPRa activity (*Figure 1—figure supplement 1A*). The CRISPRa enzymes used in this study included dCas9-VPR (*Chavez et al., 2015*) and dCas9-Vp64 (*Maeder et al., 2013*, *Figure 1—figure supplement 1B*). To drive the expression of the ECFP reporter, CRISPRa reporters containing either a single CRISPR-target sequence (1xCTS) or multiple (8xCTS) sequences were employed (*Nissim et al., 2014*, *Figure 1—figure supplement 1C*).

Upon binding of RNA triggers to complementary iSBH-sgRNA sequences, the spacer sequences become exposed inside the cells. As the RNA triggers are complementary to both the loop and the spacer* sequence, this interaction is expected to be more energetically favourable than the interaction between the spacer and spacer* components. When activated iSBH-sgRNAs bind to CRISPRa enzymes, the resulting complex can recruit transcription activation factors to the ECFP synthetic promoter, leading to the production of ECFP from the CRISPRa reporter (*Figure 1B*).

The initial test performed using dCas9-VPR as well as 1xCTS-ECFP reporters showed that iSBH-sgRNA could be activated by RNA triggers, and the observed ON-state activation was comparable to the activation seen in cells transfected with native sgRNAs (spacer only, no iSBH fold, *Figure 1—figure supplement 1D*). In comparison, for all five iSBH-sgRNA sequences, a closed hairpin (OFF-state, absence of RNA trigger) significantly reduced CRISPRa activity, as demonstrated by a highly reduced reporter expression.

Although iSBH-sgRNAs reduced CRISPRa activity, background activation levels were still detectable in the OFF-state. This could be due to a percentage of iSBH-sgRNAs molecules that might not adopt desired secondary structures and stronger activators such as dCas9-VPR (*Chavez et al., 2015*) may then propagate this background noise. We reasoned that weaker activators such as dCas9-Vp64 (*Maeder et al., 2013*), which require concomitant binding of several effectors at the promoter to efficiently drive downstream gene expression, could mask this noise. Therefore, we tested our system using dCas9-Vp64 (*Maeder et al., 2013*) in combination with the 8xCTS-ECFP (*Nissim et al., 2014*) reporter cassette (*Figure 1C*). Changing these components not only reduced the OFF-state activation but also increased the intensity of the ECFP signal detected by Flow Cytometry in the ON-state (*Figure 1—figure supplement 1E*). Unless specified otherwise, all following work and ensuing figures were generated using dCas9-Vp64 and 8xCTS reporters.

To further verify whether iSBH-sgRNA activation is specific, we tested activation for all 25 iSBH-sgRNAs/RNA trigger combinations (*Figure 1D*). Data showed that iSBH-sgRNAs only become activated in the presence of their corresponding triggers, suggesting the exquisite activation specificity of our system. Furthermore, orthogonality between different iSBH-sgRNA and their triggers suggested that multiple iSBH-sgRNA trigger pairs could be incorporated within genetic circuits and used in parallel for performing different tasks.

## Design optimisations enable CRISPR activation in the presence of longer RNAs

As most biologically relevant RNA sequences are longer than the 34 nt RNA triggers detected by first-generation iSBH-sgRNAs (*Figure 1*), we next sought to detect specific RNA sequences embedded within longer transcripts (*Figure 2*). Previous studies suggested that extending the length of hybridisation probes improved RNA targeting (*Qu et al., 2019*; *Hasegawa et al., 2006*). Starting from this principle, second-generation iSBH-sgRNAs were designed by extending the length of the iSBH-sgRNA backfold complementary with the RNA trigger. A 10 nucleotide (nt) extension was introduced between the spacer and loop sequences, resulting in a 30 nt backfold. Increasing the size of iSBH-sgRNAs enabled increasing the size of complementary RNA triggers from 34 to 44 nt (*Figure 2—figure supplement 1A*).

In an initial experiment, we tested if second-generation iSBH-sgRNA designs were silent in an OFF-state and could still detect short RNA triggers. The performance of second-generation iSBH-sgRNA designs was tested using six different combinations of CRISPRa reporter systems (*Figure 2—figure supplement 1*). Similar to the assessment performed for first-generation designs, we used 1xCTS-ECFP and 8xCTS-ECFP reporters. In terms of CRISPRa effectors, dCas9-Vp64 was included as a weak activator and dCas9-VPR as a stronger activator. Furthermore, dCas9-Vp64 was also tested in conjunction with the synergistic activation mediator (SAM) amplification system (*Konermann et al., 2015*).

In concordance with our observations for first-generation iSBH-sgRNA designs, dCas9-Vp64 and 8xCTS-ECFP had the cleanest OFF-state (*Figure 2—figure supplement 1E*). Nevertheless, combinations between the 8xCTS-ECFP reporter and dCas9-VPR (*Figure 2—figure supplement 1F*) or dCas9-Vp64 and the SAM system (*Figure 2—figure supplement 1G*) substantially improved the ON-state activation, but compromised a clean OFF-state. These findings further support the hypothesis that the noise resulting from a portion of iSBH-sgRNA molecules that may not adopt desired secondary structures gets either amplified or masked by strong or weak CRISPRa activator/reporter combinations, respectively.

Next, we compared the ability of the first- and second-generation iSBH-sgRNAs to detect longer RNA triggers (*Figure 2A and B*). First, we designed longer RNA triggers by appending a 100 nt flank to the 3' end of short RNA triggers (100 nt 3' flanks). For first-generation designs, activation in the presence of the trigger with 100 nt 3' flank was efficient only for iSBH-sgRNA 1 (*Figure 2C*). For second-generation iSBH-sgRNA designs, notable activation was observed for all three iSBH-sgRNAs tested, with ECFP production detected in 15–30% of the transfected cells. This result confirmed that second-generation iSBH-sgRNA designs have superior abilities in detecting longer RNA triggers. Next, we tested whether second-generation iSBH-sgRNAs could also detect other long RNA trigger designs (*Figure 2B*) and appended 100 nt flanks at the 5' end (100 nt 5' flank) or at both ends (100 nt 5'+3' flanks) of the original 44 nt trigger sequence. Second-generation iSBH-sgRNAs successfully detected different RNA trigger configurations, including triggers with sizes exceeding 250 nt (*Figure 2D*).

## Computational pipeline for custom iSBH-sgRNA design

In the second-generation iSBH-sgRNAs, the requirement for complementarity between RNA trigger and the iSBH-sgRNA backfold resulted in the restriction of RNA trigger sequence choices by the spacer sequences. As such, these designs do not allow users to detect any RNA triggers of choice while targeting Cas9 to the desired CRISPR-targeting sequence (CTS). We reasoned that the detection of biologically relevant RNA sequences would be largely facilitated by developing a platform for designing modular iSBH-sgRNAs where spacer and trigger-sensing components could be chosen independently (*Figure 3A*).

A way of achieving this involved reducing the extent of complementarity between the RNA trigger and iSBH-sgRNAs (*Figure 3—figure supplement 1A*), as well as between spacer sequences and CTSs (*Figure 3—figure supplement 1B*). We thus conceived a modular iSBH-sgRNA prototype in which RNA triggers are only complementary to the loop and the first 15 nt of the iSBH-sgRNA backfold, and iSBH-sgRNAs have only 17 nt spacer sequences complementary with CTSs. To mitigate the impact of truncating RNA trigger sequences on the affinity between iSBH-sgRNAs and RNA triggers, we increased the size of iSBH-sgRNA loops (*Figure 3A*).

Starting from the modular iSBH-sgRNA prototype, we developed the MODesign computational pipeline for enabling iSBH-sgRNA design using input RNA triggers, sgRNA spacers, and desired loop

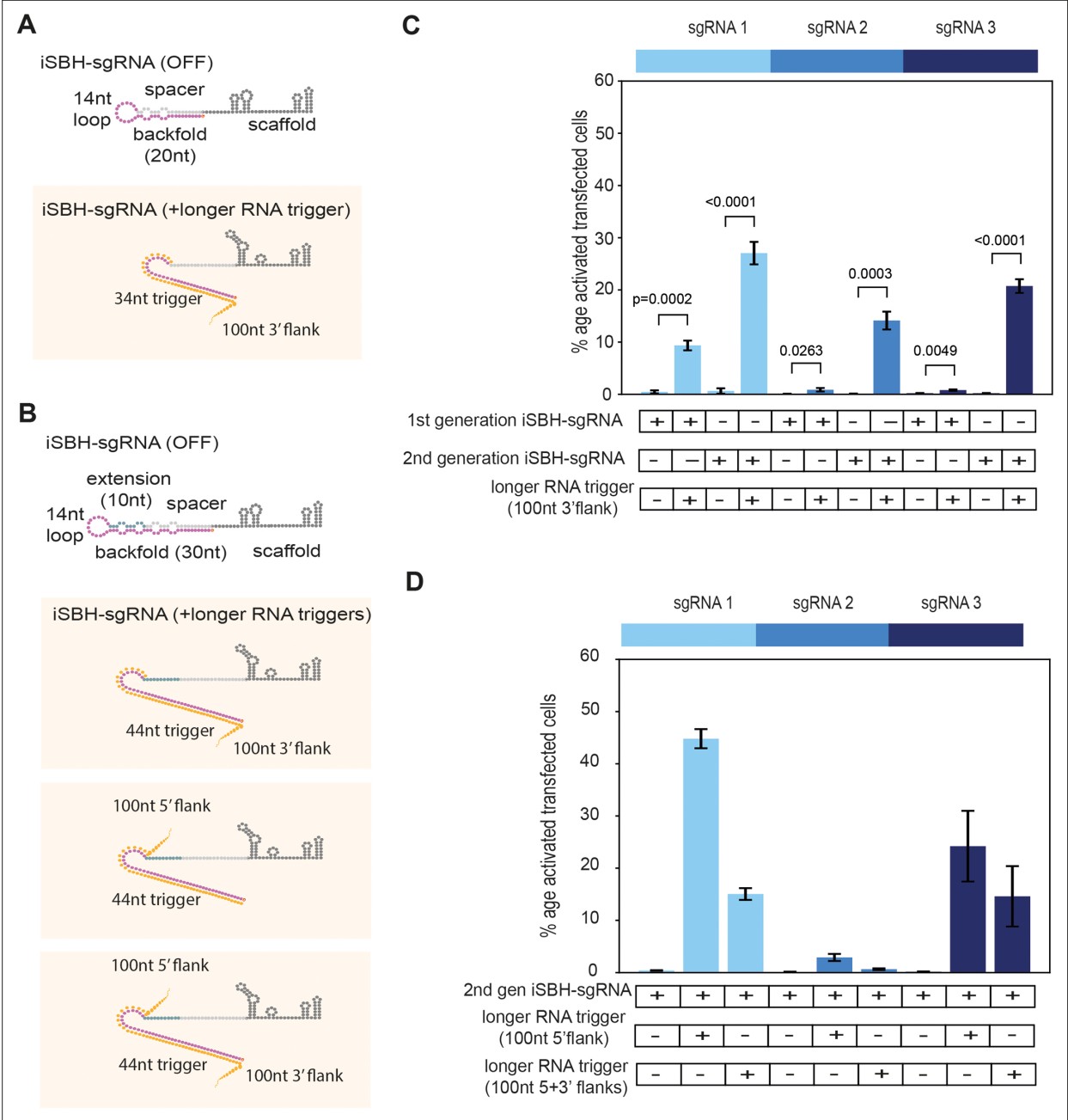

**Figure 2.** Second-generation iSBH-sgRNAs detect longer RNA triggers in HEK293T cells. (**A**) Longer RNA triggers complementary with first-generation iSBH-sgRNAs have a 34 nt sequence complementary with the loop and spacer* iSBH-sgRNA sequences. Triggers also have a 100 nt flanking sequence immediately downstream from the iSBH-sgRNA complementary region. (**B**) Second-generation designs contain a longer hairpin structure. A 10 nt extension region was inserted between the spacer and loop sequences. This enabled increasing the size of the backfold sequence to 30 nt. Longer RNA triggers complementary with second-generation iSBH-sgRNAs were designed, including triggers with 100 nt 3' flanks, 100 nt 5' flanks as well as 100 nt 5' and 3' flanks. All trigger designs contain 44 nt sequences complementary with the loops and backfold of the second-generation iSBH-sgRNAs. (**C**) Ability of first-generation and second-generation iSBH-sgRNAs to sense 100 nt 3' flank triggers was assessed. (**D**) Ability of second-generation iSBH-sgRNAs to detect different triggers with 100 nt 5' flanks and 100 nt 5' and 3' flanks was assessed. Figure shows mean ± standard deviation values measured for three biological replicates. Values above bars represent fold turn-on values for iSBH-sgRNA activation (blue) and p-values (black) determined through unpaired *t*-tests.

The online version of this article includes the following figure supplement(s) for figure 2:

**Figure supplement 1.** CRISPRa reporters of choice influence ON/OFF ratios of second generation iSBH-sgRNA designs while detecting short RNA triggers.

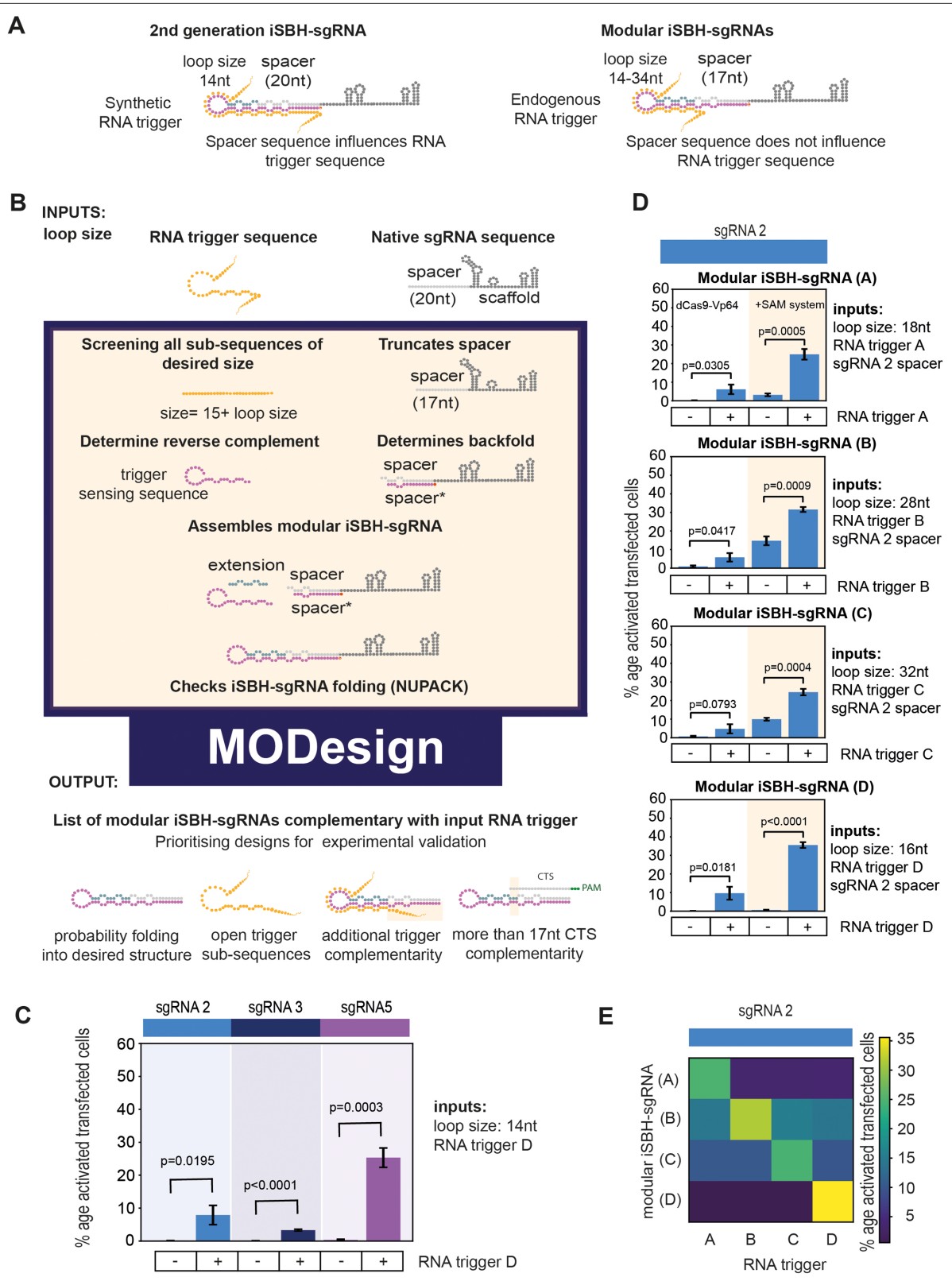

**Figure 3.** Modular iSBH-sgRNA designs enable spatial separation of spacer and trigger-sensing sequences. (**A**) In second-generation iSBH-sgRNAs, RNA triggers are complementary with the iSBH-sgRNA backfolds, thus sgRNA spacers influence RNA trigger sequences. In modular iSBH-sgRNAs, design constraints were eliminated as triggers are only complementary with the iSBH-sgRNA loop and first 15 nt of the backfold. To increase affinity between iSBH-sgRNAs and RNA triggers, we increased loop sizes. Separation between trigger-sensing and spacer sequences was also achieved by

*Figure 3 continued on next page*

*Figure 3 continued*

reducing the complementary between the spacer sequence and CTS from 20 to 17 nt. (**B**) MODesign enables users to design modular iSBH-sgRNAs starting from input RNA triggers, sgRNA spacers, and loop sizes. MODesign calculates the size of trigger-sensing sequences and creates a list of trigger sub-sequences having that size. Script determines the reverse complement of these sequences that could act as trigger-sensing sequences. iSBH-sgRNAs are assembled through adding spacer*, trigger-sensing sequences, extension, spacer, and scaffold sequences. Extension sequences are engineered to be partially complementary with trigger-sensing sequences. Before producing a list of output sequences, iSBH-sgRNA folding is checked using NuPACK (*Allouche, 2011*). Simulations could result in multiple modular iSBH-sgRNA designs. Designs chosen for experimental validation were selected based on the probability of folding into the iSBH-sgRNA structure and lack of trigger secondary structures in the iSBH-sgRNA complementary region. Priority was also given to iSBH-sgRNAs that, by chance, displayed extra complementarity between RNA triggers and the last 15 nt of the backfold or more than 17 nt complementarity with the CTS. (**C**) MODesign simulations were carried out for designing iSBH-sgRNAs capable of sensing trigger RNA D (146 nt eRNA sequence). In each simulation, a different sgRNA sequence was used and a desired loop size of 14 nt was kept constant between simulations. Selected designs were transfected to HEK293T cells together with the RNA trigger D sequence (expressed from a U6 promoter). Tests were carried out using dCas9-Vp64 and 8xCTS-ECFP reporters. (**D**) MODesign simulations were run for designing iSBH-sgRNAs capable of sensing trigger RNA A (146 nt repetitive RNA sequence), trigger RNA B (267 nt repetitive RNA sequence), trigger RNA C (268 nt repetitive RNA sequence), and trigger RNA D (146 nt eRNA sequence). Tests were performed using different CRISPRa effectors. (**E**) Four modular iSBH-sgRNAs (A–D) were co-transfected to HEK293T cells and all iSBH-sgRNA: RNA trigger combinations were tested. Figure shows mean ± standard deviation values measured for three biological replicates. Values above bars represent fold turn-on values for iSBH-sgRNA activation (blue) and p-values (black) determined through unpaired *t*-tests.

The online version of this article includes the following figure supplement(s) for figure 3:

**Figure supplement 1.** Modular iSBH-sgRNA designs enable spatial separation of spacer and trigger-sensing sequences.

**Figure supplement 2.** This supplementary figure reinterprets the data presented in *Figure 3E* using bar plots for enhanced clarity and comparison.

sizes (*Pelea, 2025a*; *Figure 3B*). MODesign calculates the size of the iSBH-sgRNA trigger-sensing component and creates a list of all potential trigger sub-sequences having that particular size. For each sub-sequence, it determines the reverse complementary region and inserts it between extension and spacer* sequences while filling in the extension sequence. Next, MODesign verifies whether resulting modular iSBH-sgRNA sequences fold into desired RNA structures using NuPACK (*Allouche, 2011*) and outputs all designs that adopt a correct fold.

MODesign simulations produce a list of multiple modular iSBH-sgRNA outputs depending on input parameters. For proof-of-concept experiments, we decided to make educated guesses about which iSBH-sgRNAs to select for experimental validation (*Figure 3B*). The first criterion was to prioritise sequences predicted by NuPACK to adopt desired structures with high probabilities. We reasoned that a higher probability of adopting desired secondary structures would reduce the number of iSBH-sgRNAs that do not adopt perfect OFF switches within the cellular pool. Priority was also given to sequences that hybridise to RNA trigger sub-sequences lacking complex secondary structures.

In the modular iSBH-sgRNA design, triggers were designed to base-pair only with the iSBH-sgRNA loop and the first 15 nt of the backfold, while iSBH-sgRNA spacers only had 17 nt complementarity with the CTS (*Figure 3A*). Nevertheless, by random chance, depending on the trigger sensing sequence, modular iSBH-sgRNAs could have extra complementarity between RNA triggers and the last 15 nt of the backfold or more than 17 nt complementarity with the CTS. As these features are beneficial for iSBH-sgRNA activation and for detecting CRISPRa activity (*Figure 3—figure supplement 1A and B*), extra priority was also given to sequences displaying these features.

In a first validation experiment, three MODesign simulations were run for designing iSBH-sgRNAs capable of sensing a U6-driven RNA trigger whose sequence corresponded to a 146 nt mouse α-globin enhancer RNA (trigger D). Initial modular iSBH-sgRNAs had different sgRNA spacer sequences and 14 nt loops. Outputs resulting from MODesign were cloned and co-transfected into HEK293T cells with plasmids expressing trigger D from U6 promoters. Our results suggested that the modular designs generated by the MODesign algorithm had good OFF-state activity and were activated by RNA trigger D (*Figure 3C*). These tests were carried out using a combination of a 'weaker' dCas9-Vp64 activator and 8xCTS-ECFP reporters.

After recognising that modular iSBH-sgRNAs can be designed starting from different sgRNA spacer sequences, a second test was performed to investigate whether these designs could detect different input triggers. Trigger RNA A (146 nt), trigger RNA B (267 nt), trigger RNA C (268 nt), and trigger RNA D (146 nt) were chosen for this purpose. Triggers A, B, and C involved a mix of zebrafish enhancer RNAs and repetitive element sequences specifically upregulated in the neural crest (*Trinh et al., 2017*). For mammalian cell tests, these triggers were expressed under the control

of U6 promoters. We ran 11 MODesign simulations for each trigger, incrementally extending the loop size while keeping the sgRNA 2 spacer input constant. HEK293T validation experiments showed that choosing modular iSBH-sgRNAs that detect the four U6-expressed triggers is possible (*Figure 3D*, *Figure 3—figure supplement 1C*). Despite not performing quite as well as second-generation designs (*Figures 2A and 3D*), modular iSBH-sgRNA still enable efficient RNA detection, especially for smaller RNAs such as triggers A and D. For highly efficient designs such as modular iSBH-sgRNA (D), addition of the SAM effector system (*Konermann et al., 2015*) boosted ON-state activation with only a negligible increase in the OFF-state non-specific activation. Orthogonality tests suggested that activation of modular iSBH-sgRNA designs was specifically conditioned by complementary RNA triggers (*Figure 3E*, *Figure 3—figure supplement 2*), showing the exquisite specificity of the system.

## iSBH-sgRNA activation occurs through RNA cleavage

Next, we sought to investigate the mechanisms of iSBH-sgRNA activation to further benefit iSBH-sgRNA technology development. It is known that in eukaryotic cells double-stranded RNAs are recognised and cleaved by endogenous RNA processing pathways such as RNA interference (RNAi, *Meister and Tuschl, 2004*; *Pong and Gullerova, 2018*). Furthermore, the interaction between iSBH-sgRNAs and RNA triggers leads to the formation of long double-stranded RNA structures and similar structures were reported to act as a non-canonical substrate for Dicer (*Pong and Gullerova, 2018*; *Burger et al., 2017*). Therefore, we tested whether double-stranded RNA processing mechanisms occur during delivery and subsequent activation of iSBH-sgRNAs.

We proposed two scenarios for the iSBH-sgRNA activation (*Figure 4A*). In the first scenario, activation would happen due to RNA strand displacement, and resulting RNA duplexes would not be processed. If this mechanism were true, the sizes of the iSBH-sgRNA and RNA triggers would be expected to remain constant after activation. A second scenario would involve double-stranded RNA processing. As a consequence, iSBH-sgRNAs and RNA triggers would be truncated following activation. To test these scenarios, we carried out RNA circularisation assays (*Knapp et al., 2019*) to measure the size of iSBH-sgRNAs (*Figure 4B*) and RNA triggers (*Figure 4D*) following activation.

In the first instance, we assessed the size of second-generation iSBH-sgRNA designs in the presence or absence of short, complementary RNA triggers and dCas9-Vp64 (*Figure 4C*). In the absence of RNA triggers, recovered iSBH-sgRNA RT-PCR bands matched the expected 137 bp size. In the presence of dCas9-Vp64 and complementary RNA triggers, the band sizes decreased to 81 bp, similar to bands recovered from non-engineered native sgRNA control. These results suggested that engineered sgRNA components (extension, loop, backfold) were removed from iSBH-sgRNAs during activation, and this hypothesis was confirmed by sequencing PCR products recovered from truncated samples (*Figure 4—figure supplement 1A*).

In the absence of dCas9-Vp64, out of 3 iSBH-sgRNAs tested, only iSBH-sgRNA 1 gets efficiently truncated (*Figure 4B*). Nevertheless, the truncated product's size is larger than that of the truncated products recovered in the presence of dCas9. These results suggested that the formation of sgRNAs with 20 nt spacers may be dCas9-dependent. In the OFF-state, iSBH-sgRNAs were unable to bind to dCas9. However, when RNA triggers were present, the conformational change enabled iSBH-sgRNAs to bind to dCas9, leading to iSBH-sgRNA truncation. These results are also supported by previous studies reporting that engineered sgRNAs with extended spacer sequences are processed to original 20 nt spacer lengths (*Perli et al., 2016*; *Ran et al., 2013a*).

We next explored the fate of RNA triggers during activation and measured the sizes of longer RNA triggers with 100 nt 3' flanks in various experimental conditions. The results demonstrated that RNA triggers are truncated specifically in the presence of matching iSBH-sgRNAs. When co-transfected together with iSBH-sgRNAs with incompatible hairpins, RNA triggers remained intact. Interestingly, unlike iSBH-sgRNAs, truncation of RNA triggers was not dCas9-dependent (*Figure 4E*). Sequencing results confirmed deletions in the truncated RNA triggers that overlapped with the 44 nt sequence complementary with iSBH-sgRNAs (*Figure 4—figure supplement 1B*).

To sum up, RNA circularisation assays suggested the involvement of double-stranded RNA processing mechanisms when iSBH-sgRNAs are deployed. Interestingly, dCas9 also seems involved in truncating iSBH-sgRNA sequences in the presence of complementary RNA triggers.

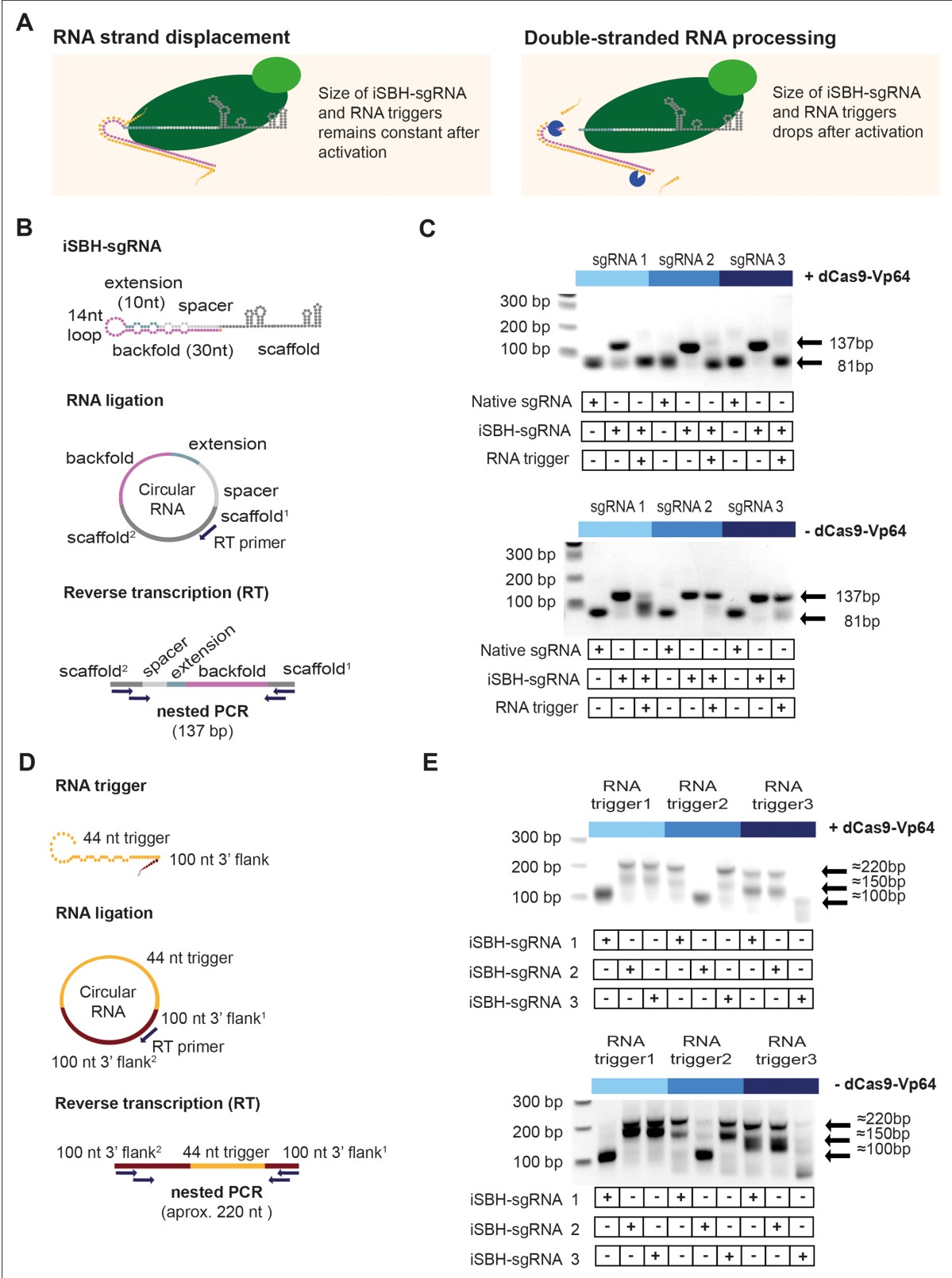

**Figure 4.** Insights into the mechanism of iSBH-sgRNA activation. (**A**) Interaction between iSBH-sgRNAs and RNA trigger leads to the formation of long double-stranded RNA structures. A potential activation mechanism might involve RNA strand displacement and formation of stable molecular complexes between the iSBH-sgRNA and the RNA trigger sequence. Supposing this scenario is correct, the size of the iSBH-sgRNA and RNA triggers is expected to remain constant after activation. A second scenario involves double-stranded RNA processing. If this is correct, iSBH-sgRNAs and RNA

*Figure 4 continued on next page*

*Figure 4 continued*

trigger sequences are expected to be truncated. (**B**) iSBH-sgRNA circularisation assay. Cells were transfected with system components, followed by RNA extraction and ligation. Reverse transcription (RT) was performed on circular RNAs by using RT primers complementary with the sgRNA scaffold. The size of the RT products was determined by two sequential PCR reactions. PCR primers annealed with the scaffold[2] and scaffold[1] sequences, which are the scaffold sequences found downstream and upstream from the RT primer. For a full-length iSBH-sgRNA sequence, a second PCR product of 137 bp is expected, while for a non-engineered native sgRNA, an 81 nt product is expected. (**C**) Determining the size of the iSBH-sgRNA after activation. Assays were performed in the presence or absence of complementary 44 nt, short RNA triggers and dCas9-Vp64. Non-engineered, native sgRNA controls were also included. (**D**) RNA trigger circularisation assay. After transfection, RNA extraction, and RT, RNA trigger size was determined by nested PCR. PCR primers annealed with the 100 nt 3' flank[2] and 100 nt 3' flank[1] sequences, which are the flank sequences downstream and upstream from the RT primer. For full-length RNA triggers, 220 bp PCR bands are expected. (**E**) Determining the size of the RNA triggers after activation. Assays were performed in the presence or absence of a complementary iSBH-sgRNAs and dCas9-Vp64.

The online version of this article includes the following source data and figure supplement(s) for figure 4:

**Source data 1.** Uncropped raw gels for *Figure 4C and E*.

**Source data 2.** Uncropped gels with relevant bands labelled for *Figure 4C and E*.

**Figure supplement 1.** Sequencing results for iSBH-sgRNA circularisation assays.

## iSBH-sgRNAs enable modulation of CRISPR activity in vivo

We next decided to test the ability of iSBH-sgRNAs to detect RNA triggers in vivo. Due to their relative ease of manipulation, transparency, small size, and key roles in developmental studies, we opted to use zebrafish embryos. We generated two transgenic zebrafish lines encoding dCas9-Vp64 and the 8xCTS-ECFP reporter recognised by one of our best-performing sgRNAs tested in mammalian cells (*Figure 5A*). An initial experiment involved testing if transgenic lines expressed ECFP in the presence of non-engineered, native sgRNAs (*Figure 5—figure supplement 1*). Initial optimisations were required to ensure homogeneous sgRNA delivery across embryo tissues (*Figure 5—figure supplement 1B–D*). After multiple optimisation steps, injection of native sgRNAs with chemical modifications resulted in optimal ECFP activation across tissues. Furthermore, ECFP activation persisted several days post-injection (*Figure 5—figure supplement 1D*).

Then, iSBH-sgRNAs were injected into transgenic embryos in the absence or presence of complementary RNA triggers. In the absence of RNA triggers, embryos are expected to be ECFP negative, while in the presence of complementary RNA triggers, embryos should express ECFP (*Figure 5B*). Due to improved sgRNA activity in the presence of chemical modifications, we then tested iSBH-sgRNAs with chemical modifications. A first iteration of chemically modified iSBH-sgRNAs was designed to protect the 5' end of the iSBH-sgRNA and the sgRNA scaffold. However, when co-injected with chemically synthesised RNA triggers, these modifications did not lead to activation of the 8xCTS-ECFP reporter (*Figure 5—figure supplement 2A*). Our mechanistic data suggested that iSBH-sgRNAs get truncated during activation (*Figure 4*) and, in a first iteration of chemically modified iSBH-sgRNAs, such truncation would lead to a production of sgRNAs with unprotected spacer sequences. In this scenario, 5'–3' sgRNA degradation was likely to occur. We thus reasoned that additional chemical modifications in the spacer sequence would reduce degradation by cellular nucleases and promote iSBH-sgRNA function in vivo. Therefore, we designed a second strategy for protecting iSBH-sgRNA integrity through chemical modifications (*Figure 5C*). This was achieved by simultaneously protecting the 5' end of the iSBH-sgRNA, the 5' end of the spacer, and the scaffold sequences.

iSBH-sgRNAs synthesised using our second chemical modification strategy (*Figure 5C*) were injected in the absence or presence of chemically synthesised RNA triggers (*Figure 5B*), and stronger ECFP signals were detected in the presence of matching RNA triggers (iSBH-sgRNA ON, *Figure 5—figure supplement 2B*). To quantify ECFP activation, we grouped fish into three categories, according to their level of ECFP expression (no ECFP, low ECFP, and high ECFP, *Figure 5D*). $\chi^2$ tests (*Figure 5E*) performed for three experimental replicates showed that the presence of RNA triggers causes a statistically significant change in ECFP expression. We then plotted percentages of embryos counted for each category (*Figure 5F*). Data shows an increase in the number of embryos with high ECFP levels in the presence of RNA triggers. In contrast with the iSBH-sgRNA (OFF) condition, the iSBH-sgRNA (ON) condition presented an average of 3.2-fold increase in the number of embryos with high ECFP signals. Furthermore, the average number of embryos with no ECFP signals recovered in the iSBH-sgRNA (ON) condition was lower than in the iSBH-sgRNA (OFF) condition.

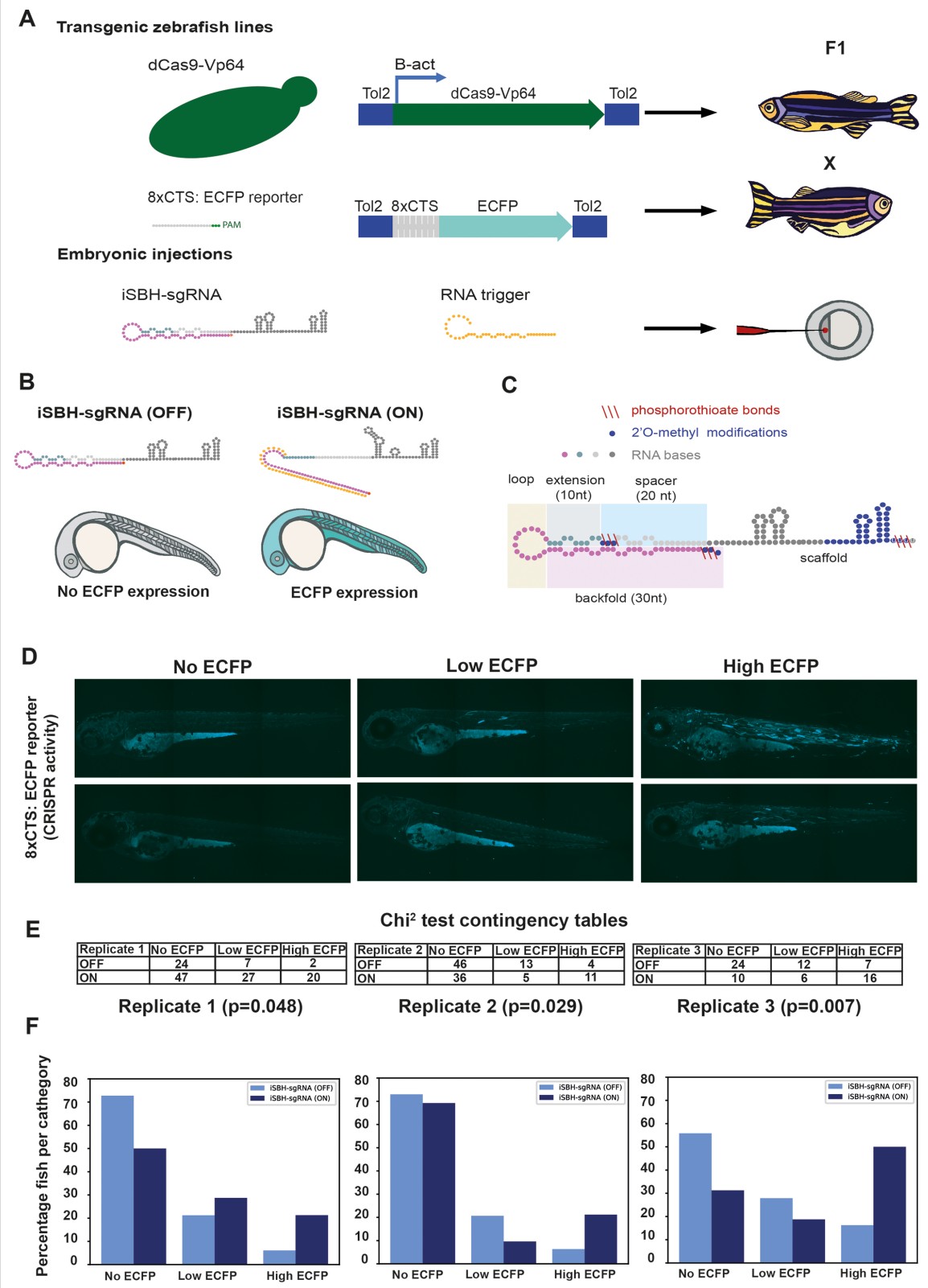

**Figure 5.** Testing the ability of second-generation iSBH-sgRNA designs to detect short RNA triggers in vivo. (**A**) Transgenic lines encoding dCas9-Vp64 and 8xCTS-ECFP reporters were created. Embryos resulting from in-crossing first-generation (F1) transgenics were injected with second-generation chemically synthesised iSBH-sgRNAs and RNA triggers. (**B**) Second-generation iSBH-sgRNAs were injected into transgenic zebrafish embryos with or without corresponding short RNA triggers. In the absence of RNA triggers (iSBH-sgRNA OFF), embryos are expected to display no ECFP signals, while

*Figure 5 continued on next page*

*Figure 5 continued*

trigger presence (iSBH-sgRNA ON) should promote ECFP expression. (**C**) Figure presents our second strategy for chemically modifying iSBH-sgRNAs. This strategy involved protecting the iSBH-sgRNA 5′ end as well as the 5′ end of the sgRNA spacer. These modifications were used together with sgRNA scaffold modifications. (**D**) In order to quantify the impact of RNA triggers on iSBH-sgRNA activation, we grouped fish according to the intensity of ECFP signals. At 3 days post-fertilisation, embryos displaying no, low, or high ECFP expression were counted. (**E**) Embryos injected with iSBH-sgRNAs and non-complementary (iSBH-sgRNA OFF) of complementary RNA triggers (iSBH-sgRNA ON) were scored according to their ECFP intensity. Row number counts determined for three experimental replicates are displayed as part of $\chi^2$ contingency tables. p-Values displayed were determined using $\chi^2$ test. (**F**) Figure shows percentage of embryos recovered in each category for the three experimental replicates. Percentage of embryos with no ECFP expression varied between the three experimental replicates. This was due to the fact that both 8xCTS-ECFP and dCas9-VP64 transgenes are necessary for successfully expressing ECFP. These alleles segregate in a Mendelian fashion and our adult transgenic fish encode variable copy numbers of the transgene. For each individual replicate, we used embryos with identical genetic backgrounds for testing the iSBH-sgRNA (OFF) and iSBH-sgRNA (ON) conditions. Nevertheless, genetic backgrounds were different between the three experimental replicates.

The online version of this article includes the following figure supplement(s) for figure 5:

**Figure supplement 1.** Optimising sgRNA delivery to zebrafish embryos.

**Figure supplement 2.** Testing different iSBH-sgRNA chemical modifications in vivo.

Our results show that iSBH-sgRNAs are functional in developing zebrafish embryos. Furthermore, gaining insight into the mechanism of the iSBH-sgRNA activation has allowed us to identify critical positions for chemical modifications that stabilised our design.

## Discussion

In this study, we provide evidence that iSBH-sgRNAs enable conditional CRISPR activation in response to RNA detection in both HEK293T cells and zebrafish embryos. We adopted a Synthetic Biology design-build-test cycle to develop iSBH-sgRNAs that can detect longer RNA triggers. We also developed the MODesign algorithm, which allows users to design modular iSBH-sgRNAs that detect desired RNA triggers while directing the CRISPR activator to a gene target of choice. We successfully detected RNA trigger sequences of up to 300 nt expressed from U6 promoters in HEK293T cells using modular iSBH-sgRNA designs. Orthogonality tests suggest that modular iSBH-sgRNA designs are highly specific to their trigger and can be custom-made for different RNA detection applications. Our data show that the choice of CRISPR activators and reporters greatly influences the dynamic ranges of iSBH-sgRNA activation. Furthermore, we also provide insights into the iSBH-sgRNA activation mechanism. Our results suggest that dCas9 is involved in iSBH-sgRNA processing and activation. iSBH-sgRNA truncation to sgRNA sequences that have 20 nt spacers is consistent with previous observations reported in studies that attempt to extend spacer sequences (*Perli et al., 2016*; *Ran et al., 2013a*). Our data suggest that RNA triggers are cleaved by endogenous factors and molecules from the RNA interference pathway (*Meister and Tuschl, 2004*; *Pong and Gullerova, 2018*) could be responsible for the cleavage.

To date, a variety of RNA-inducible gRNA designs have been developed (*Hanewich-Hollatz et al., 2019*; *Hochrein et al., 2021*; *Jakimo et al., 2018*; *Jiao et al., 2021*; *Jin et al., 2019*; *Li et al., 2019*; *Liu et al., 2022*; *Lin et al., 2020*; *Siu and Chen, 2019*; *Galizi et al., 2020*; *Hunt and Chen, 2022a*; *Hunt and Chen, 2022b*; *Ying et al., 2020*; *Choi et al., 2023*). Nevertheless, there is a lack of direct, head-to-head comparisons of these designs under standardised experimental conditions. Some designs were evaluated in vitro, others in bacterial systems, and some in mammalian cells. Consequently, it is challenging to conclusively determine which design exhibits superior properties (*Pelea et al., 2022*). Notably, to the best of our knowledge, the iSBH-sgRNA system is the first RNA-inducible gRNA design tested in vivo and characterising the iSBH-sgRNA activation mechanism was essential for implementing iSBH-sgRNA technology in zebrafish embryos. In vivo, chemical modifications in the spacer sequence were vital for iSBH-sgRNA stability and function.

In their current iteration, iSBH-sgRNAs show considerable promise for mammalian synthetic biology applications. Specifically, their ability to detect synthetic triggers could be pivotal in the development of complex synthetic RNA circuits and logic gates, thereby advancing the field of cellular reprogramming. However, further work is required to achieve better ON/OFF activation ratios in vivo and more homogeneous activity across tissues in the presence of RNA triggers. Additional chemical modifications could improve iSBH-sgRNA properties, and we believe that chemical modification strategies

adopted for siRNA drugs or antisense oligos (*Khvorova and Watts, 2017*) could also be essential for further iSBH-sgRNA technology development. As iSBH-sgRNAs might be targeted by endogenous nucleases, leading to their degradation, a strategy for preventing this could involve additional chemical modifications. When inserted at certain key positions, such modifications could prevent interaction between iSBH-sgRNAs and cellular enzymes by introducing steric clashes or inhibiting RNA hydrolysis.

Once achieving superior dynamic ranges of iSBH-sgRNA activation in vivo, the next steps would involve understanding the classes of endogenous RNAs that could act as triggers. The chances that an iSBH-sgRNA encounters an endogenous RNA trigger inside a cell would depend on the relative concentrations of the two RNA species. Therefore, a first step towards determining potential endogenous RNA triggers will involve identifying RNA species with comparable expression levels as iSBH-sgRNAs. Then, iSBH-sgRNAs could be designed against these RNA species, followed by experimental validation. It is important to note that eukaryotic cells express a wide range of transcripts of varying sizes, expression levels, and subcellular localisations, all of which could greatly affect iSBH-sgRNA activation levels. Based on the data presented here, we speculate that RNA species up to 300 nt that are also highly expressed might act as good triggers. Furthermore, as sgRNAs are involved in targeting Cas9 to genomic DNA in the nucleus, attempting to detect transcripts that are sequestered in the nucleus might also provide additional benefit.

After identifying RNA species that could act as triggers in vivo, iSBH-sgRNAs could pave the way for the development of more effective gene editing approaches with greater specificity and efficiency. In the field of therapeutics, safety concerns regarding the off-target effects of CRISPR-Cas9 persist. CRISPR-Cas9 commonly results in off-target effects such as Cas9 deployment at unintended genomic regions and the induction of unwanted DNA double-stranded breaks (DSBs, *Wu et al., 2014*). These DSBs can trigger chromosomal rearrangements and macro deletions (*Kosicki et al., 2018*) and activate a p53-mediated DNA damage response (*Haapaniemi et al., 2018*). Thus, it is crucial to restrict the activity of CRISPR components to the affected tissues to prevent off-target effects in healthy tissues that are not impacted by the disease (*Doudna, 2020*). Recent developments in engineering lipid nanoparticles have demonstrated selective accumulation in different target organs (*Cheng et al., 2020*; *Rosenblum et al., 2020*; *Wei et al., 2020*). However, targeted delivery strategies still rely heavily on the availability of cell-surface protein biomarkers (*Rosenblum et al., 2020*; *Dilliard et al., 2021*), such as membrane receptors, which act as a proxy for cell identity. The RNA-sensing iSBH-sgRNA technology could provide a complementary approach to achieve targeted delivery by sensing endogenous RNA biomarkers expressed specifically in the affected tissues. In the pursuit of targeted gene editing, identifying cell surface biomarkers can be a daunting task for certain cell types or diseases. However, a promising solution may lie in utilising endogenous RNA biomarkers to activate CRISPR activity (*Lee et al., 2019*). Rather than relying on targeted delivery methods, iSBH-sgRNAs can be delivered to cells in an inactive form, only to be activated upon detection of specific RNA biomarkers within the target cells. This would ensure that CRISPR-Cas9 systems remain inactive in non-target cells where these RNA biomarkers are not expressed.

## Materials and methods
### In silico design of iSBH-sgRNAs

First-generation iSBH-sgRNA designs were designed by inputting different spacer sequences into a computational pipeline generated by Ferry et al.: http://apps.molbiol.ox.ac.uk/iSBHfold/ (*Ferry et al., 2017*). Main features of iSBH-sgRNA designs include the spacer (20 nt), loop (14 nt), spacer* (20 nt), and scaffold sequences. Spacer* sequence is the reverse complement of the spacer sequence, and it was modified to contain mismatches at positions: 11–12 and 16–17. The NuPACK (*Allouche, 2011*) fold corresponding to these sequences (excluding scaffold and extra GC sequences*) is: (((((((((..(((.. (((…………..)))..)))..)))))))))); where . represents an unpaired nucleotide, while ( and ) represent paired nucleotides.

Second-generation iSBH-sgRNA designs were generated by introducing 10 nt extensions between loop and spacer sequences of the first-generation design. The random 10 nt extension sequences had a GC content ranging between 40 and 60%. Extension* sequence is generated by determining the reverse complement of the extension sequence, with mismatches integrated at positions 1–2 and

6–7. NuPACK (*Allouche, 2011*) fold corresponding to this design (excluding scaffold and extra GC sequences) is: (((((((((((..(((..((((..((((..(((…………..)))..)))..))))..)))..)))))))))))

Modular iSBH-sgRNA designs were generated using the MODesign computational pipeline (*Pelea, 2025a*). This pipeline takes three input sequences: iSBH-sgRNA loop size, the RNA trigger to be sensed an input sgRNA sequence that targets Cas9 to desired CTSs (CRISPR target sequences). MODesign calculates the size of the iSBH-sgRNA trigger-sensing component and creates a list of all potential trigger sub-sequences having that particular size. For each sub-sequence, it determines the reverse complementary region, and it inserts it between extension and spacer* sequences while filling in the extension sequence. MODesign checks if resulting modular iSBH-sgRNA sequences fold into desired RNA structures using NuPACK (*Allouche, 2011*) and outputs all designs that adopt a correct fold. NuPACK fold corresponding to this design is (excluding scaffold and extra GC sequences*): (((((((((((..(((..(((..(((..(((..(((…………..)))..)))..)))..)))..)))..)))))))))))

A MODesign scoring algorithm was also implemented, based on the iSBH-sgRNA probability of adopting desired secondary structures (N), percentage of nucleotides that are free from secondary structures in the trigger sub-sequence (M), percentage of trigger nucleotides complementary with the iSBH-sgRNA backfold (P) and percentage of spacer nucleotides complementary with the CTS (Q). The scoring formula used was $N*N*N*M*P*Q$. For different simulations, modular iSBH-sgRNAs with higher scores were selected for experimental validation.

For first-generation designs, RNA triggers complementary with the iSBH-sgRNA loop and spacer* sequences were designed. For second-generation designs, RNA triggers are complementary with the loop, extension* and spacer* sequences. CTSs were designed by adding PAM (protospacer adjacent motifs) sites downstream from the spacer sequence. All sgRNA spacers, iSBH-sgRNAs, triggers, and CTS reporter sequences could be found in the Supplementary Material.

At the beginning of each iSBH-sgRNA, an extra GC sequence was added. The extra G was inserted in order to promote transcription from the polymerase III U6 promoter (*Ran et al., 2013b*). The extra C was added in order to increase base-pairing complementary with the first G in the sgRNA scaffold.

## Molecular cloning

All mammalian plasmids expressing Cas9 and deadCas9-fused transcriptional activators were acquired from Addgene: dCas9-Vp64 (#47107), dCas9-VPR (#63798) as well as Cas9_pX458 (#48138).

Native sgRNAs, iSBH-sgRNAs, RNA triggers, and 1xCTS repeats were cloned in mammalian-expression vectors by inserting annealed single-stranded DNA oligos (IDT) into appropriate plasmid backbones. iSBH sequences, as well as sgRNA spacer sequences, were cloned between BbsI restriction sites in the pcDNA3.1-U6_sgRNA_6xT-SV40_iBue_PA plasmid, generated by *Ferry et al., 2017* sequences were cloned in p035_pause-HBG-CFP-pA (XbaI, AscI restriction sites; plasmid received as a gift from Dr. David Knapp). Short trigger sequences were cloned within U6-TM2Emp-6T_iBlue plasmid (gift from Dr. Quentin Ferry) between BbsI restriction sites. U6-TM2Emp-6T_iBlue plasmid also encoded for two hairpin structures aiming to protect 34 and 44 nt short triggers from degradation.

Forward and reverse oligos were treated with polynucleotide kinase (PNK) and incubated at 37°C for 30 minutes. Oligos were denatured and re-annealed by heating up samples to 95°C and decreasing the temperature to 25°C (at a rate of 2°C/minute). Backbone plasmids were digested according to the NEB protocols, followed by gel extraction. Ligation was carried out using 100 ng backbone and 0.5 µl annealed oligos in a total volume of 10 µl. 3 µl ligation product was transformed into DH5α *E. coli* (Invitrogen, 18265017) cells.

8xCTS sequences were cloned in the P2-ECFP-pA (Addgene #26280) plasmid generated by *Nissim et al., 2014*. The original 8xCTS sequence was removed (NheI restriction sites) and replaced with a cloning landing pad containing numerous restriction sites. Resulting vector was named Landing_Pad_8xCTS-ECFP-pA. ssDNA oligos encoding for 2xCTS repeats were ordered from IDT and amplified by PCR. PCR products were separated into two restriction reactions- SacI/Esp3I and Esp3I/SpeI, respectively. The two digestion products were cloned into the Landing_Pad_8xCTS-ECFP-pA, leading to a 4xCTS-ECFP reporter. In a subsequent round of cloning, 4xCTSs were amplified by PCR, digested with NheI/ApaI, and cloned back into the 4xCTS-ECFP reporter, leading to the final 8xCTS-ECFP constructs. Primer sequences and IDT oligos used for cloning 8xCTS reporters could be found in the Supplementary Material.

100 nt flank U6 triggers were cloned in the U6-TM2Emp-6T_iBlue_ModFlanks plasmid. In the first round of cloning, the backbone was digested using BbsI and short trigger sequences were cloned by annealing and ligation. Further rounds of cloning involved the amplification of 100 nt flanks by PCR and standard restriction-ligation cloning. 100 nt 5' flanks were digested with BsmBI and 100 nt 3' flanks were digested with NotI/NheI. The final size of triggers with 100 nt 3' flank extension was 191 nt for first-generation iSBH-sgRNA designs and 200 nt for second-generation iSBH-sgRNA designs. The size of triggers with 100 nt 5' flanks was 175 nt, while the size of triggers with 100 nt 5'+3' flanks was 270 nt. Sizes exclude two hairpin sequences that protect 5' and 3' trigger ends from degradation.

Repetitive RNA trigger sequences for testing modular iSBH-sgRNA designs were cloned in the U6-TM2Emp-6T_iBlue plasmid between BbsI restriction sites using type II S restriction-cloning. Trigger sequences were split into smaller sub-sequences and ordered from IDT as single-stranded oligos. Primers annealed to oligo sequences and contained long flaps including extra trigger sub-sequences and BbsI restriction sites. Flaps enabled the assembly of longer trigger sub-sequences by PCR. PCR products were loaded in a 1% agarose gel, followed by gel extraction, BbsI digestion and PCR clean-up. Depending on the trigger size, two or three digested PCR products were ligated to 100 ng backbone using a 3:1 molar ratio for inserts to the backbone. Examples of IDT oligos and ordered primers for cloning triggers A and B are available in the Supplementary Material.

Tol2(B-act:dCas9-Vp64-T2A-Citrine) vector was generated by Gibson Assembly. Vector backbone was amplified by PCR from pMTB2-NLS-Cas9-2a-Citrine plasmid (gift from Dr. Vanessa Chong-Morrison), while the dCas9-Vp64 cassette was amplified from the Sox10:BAC_Cas9m4-Vp64-2a-Citrine bacterial artificial chromosome (gift from Dr. Vanessa Chong-Morrison). Tol2(8xCTS-ECFP) reporter was generated by standard restriction-digestion cloning. pMTB2-NLS-Cas9-2a-Citrine was digested using MfeI and XhoI restriction enzymes (NEB). Insert was amplified by PCR and digested with BbsI (NEB) to produce sticky ends compatible with plasmid ligation.

iSBH-sgRNAs and triggers for zebrafish expression were cloned in AcDs_MiniVector_U6a-sgRNA-MS2 (gift from Dr. Vanessa Chong-Morrison). SAM effector plasmids for zebrafish expression were also modified from AcDs_MiniVector_ubb-MCP-p65-HSF1-2a-Cerulean (gift from Dr. Vanessa Chong-Morrison) by replacing Cerulean with Citrine using standard restriction-digestion cloning. AcDs vectors containing ubb-MCP-p65-HSF1-2a-Cerulean, iSBH-sgRNAs and RNA triggers were cloned using Gibson Assembly.

All PCRs were carried out using Phusion High-Fidelity PCR Master Mix with GC Buffer (NEB), gel extractions using the QIAquick Gel Extraction Kit (QIAGEN), and PCR clean-up reactions using the MinElute PCR Purification Kit (QIAGEN). DNA concentrations were estimated by NanoDrop, standard ligations were performed using T4 DNA ligase (NEB), while Gibson assembly reactions used the NEBuilder(R) HiFi DNA Assembly Cloning Kit (NEB). Transformed bacteria were grown for 24 h at 32°C, while single colonies were grown for 20 h at 37°C in ampicillin-containing LB media. Plasmid DNA was extracted using the QIAprep Spin Miniprep Kit (QIAGEN) and constructs were validated by Sanger sequencing (Eurofins Genomics) prior to being transfected into HEK293T cells or injected into zebrafish embryos.

## Maintaining HEK293T cell lines

HEK293T cells were grown in full media (Thermo Fisher Dulbecco's modified Eagle's medium supplemented with 10% foetal bovine serum). Cells were grown at 37°C and 5% $CO_2$ and passaged every 2 days. HEK293T cells (CRL-3216) were obtained from the American Type Culture Collection (ATCC), and no further cell line authentication was performed. Cells have been *Mycoplasma* tested using the Eurofins Genomics Mycoplasmacheck services.

## Transfecting HEK293T cell lines

Transfection was performed using Sigma-Aldrich polyethyleneimine (PEI). For each well, transfection DNA cocktails were mixed with 50 μl GIBCO Opti-MEM. 1.5 μl PEI/μg DNA ratio was maintained for all experiments. To each well, 250 ng Cas9 effector was transfected with 250 ng sgRNA-backbone plasmid (pcDNA3.1_SV40-iBlue-pA), 250 ng trigger-encoding plasmids and 125 ng 1xCTS or 8xCTS-ECFP reporter plasmids. For transfections involving SAM effector components, an extra 250 ng MCP-p65-HSF1-iBlue plasmid was co-transfected.

## Flow cytometry sample preparation

48 h after transfection, cells were washed with PBS and incubated for 10 min with 50µl trypsin-EDTA. Trypsin was inactivated using 250µl FACS buffer (PBS with 10% foetal bovine serum), followed by cell filtration (Falcon 70 µm White Cell Strainer) and transfer to Flow Cytometry tubes. Cells were stored on ice prior to analysis using the BD LSR Fortessa Analyzer (BD Biosciences).

## Flow cytometry data analysis

For each condition, 100,000 events were acquired, and data analysis was performed using a Python script (*Pelea, 2025b*) developed starting from the FlowCal package (*Castillo-Hair et al., 2016*). HEK293T cells were identified by plotting side-scattering area (SSC-A) and forward scattering area (FSC-A) values. Single cells were identified by plotting side-scattering height (SSC-H) against SSC-A values. Then, the script plots the iBlue (640–670 nm) transfection control reporter against ECFP (405–450 nm) reporter for monitoring CRISPR activity.

Gates for iBlue level were set up in the untransfected control, so that only 0.1% of untransfected cells are iBlue+. Gates for the ECFP control were set up in the reporter control, where the sgRNA transfected is not complementary with the nxCTS-ECFP reporter. ECFP gate was set up in such a way that around 0.1% of the cells in the reporter condition were ECFP positive. The displayed percentage of activated transfected cells was calculated using the following formula: count(ECFP+/iBlue+ cells)/ [count(ECFP+/iBlue+ cells)+count(ECFP-/iBlue+ cells)].

All bar graphs present the percentage of activated transfected cells measured for three different transfections carried out on three different days. The error bars represent the ± standard deviations for these three biological replicates. Displayed p-values were calculated using a non-paired *t*-test.

## RNA circularisation assays

RNA circularisation protocol was adapted from *Knapp et al., 2019*. 48 h after transfection, cells were washed with PBS, followed by incubation with 50 µl trypsin. Trypsin was inactivated using 500 µl FACS buffer. Cells were transferred to a 1.5 ml Eppendorf tube, followed by centrifugation at 300×$g$ for 5 min. Supernatant was removed and cells were resuspended in 500 ml PBS followed by another centrifugation step at 300×$g$ for 5 min. Supernatant was removed followed by snap-freezing of cellular pellets. Cells were stored at –80°C prior to RNA extraction using the Charge Switch Total RNA cell Kit (Thermo Fisher). All indications specified in the kit were followed, except for using 1/5 of suggested volume of buffers. The optional DNase treatment step was also included, and the RNA was eluted in 20 µl elution buffer.

The ligation reaction was set up by mixing 10 µl RNA, 2 µl T4 RNA ligase buffer (NEB), 1.9 µl H$_2$O, 4 µl 50% PEG 8000 (NEB), 0.1 µl 10 mM ATP, 1 µl T4 RNA ligase (NEB), and 1 µl SUPERase in RNAase inhibitor (Thermo Fisher). The reaction was incubated for 4 h at room temperature, followed by another round of RNA extraction using Charge Switch Total RNA cell Kit. In this second round of RNA extraction, 1/10 off specified buffer volumes were used, the optional DNase treatment was not performed, and RNA was eluted in 12.5 µl elution buffer.

10µl circular RNAs were subjected to reverse transcription (RT) using the QuantiTect Rev. Transcription Kit (QIAGEN). Manual specifications were followed, but provided RT primers were replaced with custom made primers that specifically bind to regions of interest. According to user specifications, primers were diluted to 0.7 µM and RT reactions were incubated for 30 min at 42 °C.

Desired sequences were amplified using two subsequent PCR reactions using the Phusion High-Fidelity PCR Master Mix with GC Buffer (NEB). All extension steps were carried out in a 15 s timeframe and the first PCR round consisted of 10 cycles. 2 µl RT product was used as a template in the first reaction. The product of the first reaction was diluted 1/10 and 1 µl of this dilution served as a template in the second round of PCR (25 cycles). Second PCR products were mixed with Gel Loading Dye, purple (NEB) followed by loading of 2 µl mixture into a 2% agarose gel. For optimal results, gels were run for approximately 90 min. Second PCR primers contained NotI/NheI restriction sites that enabled fragment cloning into the pcDNA3.1 plasmid followed by Sanger sequencing.

Circularisation assay primer sequences could be found in the Supplementary Material.

## Zebrafish husbandry

Zebrafish experiments were carried out according to regulated procedures authorised by the UK Home Office within the framework of the Animals (Scientific Procedures) Act 1986. Embryos used were derived from AB zebrafish strains.

## Synthesis of mRNA for embryonic injections

Templates for in vitro transcription were linearised by either restriction digestion or PCR, followed by purification using the QIAquick PCR Purification Kit (QUIAGEN). mRNA was synthesised using the mMESSAGE mMACHINE SP6 Transcription Kit (Thermo Fisher) according to the manufacturer's specifications. Following treatment with TURBO DNase, RNA was purified using the Monarch RNA Cleanup Kit (NEB). The integrity of the purified RNA was determined by running RNA in a 1% agarose gel, while RNA concentration was measured using the Qubit RNA Broad Range assay (Thermo Fisher).

## Generation of zebrafish transgenics

The transgenic lines *Tg(B-act:dCas9-Vp64-T2A-Citrine)[ox176]* (dCas9-Vp64) and *Tg(8xCTS:ECF-P)[ox178]* (8xCTS-ECFP) were generated in the background of our existing *TgBAC(Sox10:cytoBirA-2a-mCherry)[ox168]* transgenic created in *Trinh et al., 2017*. Single-cell embryos obtained by incrossing *TgBAC(Sox10:cytoBirA-2a-mCherry)[ox168]* fish were injected with DNA constructs containing Tol2 recombination arms as well as Tol2 mRNA (*Urasaki et al., 2008*). The DNA expression and reported constructs (Tol2(B-act:dCas9-Vp64-T2A-Citrine) and Tol2(8xCTS-ECFP), respectively) were built as described in the Molecular cloning subsection. Injection mixtures contained 2 µl of plasmid DNA (200 ng/µl), 1.5 µl Tol2 mRNA (160 ng/µl), as well as 0.5 µl Phenol Red. 2 nl mixture was injected using a PICOSPRITZER III injector. Following injections, embryos were kept in a 28°C incubator. 6 h post-injection, fertilised embryos were selected and transferred to E3 media. At 1 day post-fertilisation, dead embryos were removed. Surviving embryos were grown for 4 months and subsequently genotyped.

A strategy consisting of two rounds of crosses was employed for identifying founders. In a first round, potential male and female founders were incrossed for identifying founder pairs. Once a founder pair was identified, the next step was determining whether the male or the female from that cross contains the transgene. This was achieved by outcrossing males and females with wild-type fish. For each cross, DNA was extracted from a pool of embryos at 1 day post-fertilisation. Extraction was carried out using the PureLink Genomic DNA Mini kit (Thermo Fisher), while opting for a 1 h lysis step. Extracted DNA was measured by NanoDrop and 100 ng DNA was added to a first nested PCR reaction (10 cycles). Products of the first PCR reaction were diluted 1:10 and 2 µl products were transferred to a second PCR reaction (29 cycles). PCRs were carried out using the Phusion High-Fidelity PCR Master Mix with GC Buffer (NEB) in a total volume of 20 ul, while opting for an extension time of 1 min/kb. Results were assessed by running PCR products in a 2% agarose gel.

After founder identification, the next steps involved generation of the first generation of fish (F1) encoding the transgene in all cells of their body. Founders were outcrossed with TgBAC(Sox10:BirA-Cherry) adult fish. At 1–3 days post-fertilisation, embryos expressing the Sox10:BirA-mCherry transgene were selected by assessing the mCherry expression using an Olympus MVX microscope. At 3 days post-fertilisation, embryos were tail clipped. Clipped tissue was transferred to 50 µl lysis buffer (25 mM NaOH, 0.2 mM EDTA). Samples were boiled at 95°C for 45 min, followed by cooling at 4°C for 5 min. 50 µl neutralisation buffer (40 mM Tris-HCl) was added to each sample and nested PCR was carried out for determining transgene presence. PCR reactions were set up in a similar way as for founder identification, except that 5 µl DNA was added to the first PCR reaction. F1 embryos were subsequently grown for another 4 months. Subsequent experiments were carried out in embryos resulting from incrosses of F1 adult fish.

All genotyping primer sequences could be found in the Supplementary Material.

## Microinjection of zebrafish embryos using the Ac/Ds system

First generation (F1) fish encoding dCas9-Vp64 and 8xCTS-ECFP CRISPR reporter were incrossed. Resulting embryos were injected with Ac mRNA as well as plasmid DNA containing Ds transposase-recognition sequences (*Chong-Morrison et al., 2018*). Reaction mixtures contained 3 µl plasmid DNA

(266 ng/µl), 0.5 µl Ac mRNA (150 ng/µl), and 0.5 µl Phenol Red. 2 nl mixture was injected into single-cell embryos.

Following injections, embryos were kept in a 28°C incubator. 6 h post-injection, fertilised embryos were selected and transferred to E3 media. At 1 day post-fertilisation, dead embryos were removed and surviving embryos were screened for the expression of construct DNA. This was achieved by assessing Citrine expression under the Olympus MVX microscope.

## Microinjection of zebrafish embryos with chemically modified sgRNAs

F1 fish encoding dCas9-Vp64 and 8xCTS-ECFP CRISPR reporter were incrossed. Resulting embryos were injected with chemically modified sgRNAs designed and synthesised by IDT. Reaction mixtures contained 1 µl sgRNA (1 µg/µl), 2.5 µl $H_2O$, and 0.5 µl Phenol Red. 2 nl mixture was injected into single-cell embryos.

Following injections, embryos were kept in a 28°C incubator. 6 h post-injection, fertilised embryos were selected and transferred to E3 media. At 1 day post-fertilisation, dead embryos were removed. Surviving embryos were screened for ECFP production using an MVX microscope. Representative embryos were also imaged by confocal microscopy.

## Microinjection of zebrafish embryos with chemically modified iSBH-sgRNAs

F1 fish encoding dCas9-Vp64 and 8xCTS-ECFP CRISPR reporter were incrossed. Resulting embryos were injected with chemically modified iSBH-sgRNAs as well as chemically synthesised RNA triggers. iSBH-sgRNAs were co-injected together with complementary and non-complementary triggers. Reaction mixtures contained 1 µl iSBH-sgRNA (1.5 µg/ul), 1 µl RNA trigger (3 µg/ul), 1.5 µl $H_2O$, and 0.5 µl Phenol Red. 2 nl mixture was injected into single-cell embryos. 6 h post-injection, fertilised embryos were selected and transferred to E3 media. At 1 day post-fertilisation, dead embryos were removed. Surviving embryos were screened for ECFP production using an MVX microscope. For each experiment, fish were separated into three classes according to the intensity of ECFP expression: high, low, and no ECFP. Blinding was implemented by grouping embryos without knowledge of their injection condition (iSBH-OFF or iSBH-ON), ensuring unbiased assessment of fluorescence. $\chi^2$ tests were performed for testing if results are statistically significant.

## Confocal microscopy

Embryos were anaesthetised in MS222 and mounted in 1% low-melting point agarose (Invitrogen) dissolved in E3 media. Embryos were imaged on a Zeiss780 LSM upright confocal microscope using a ×10 objective. For DNA-based injections, Citrine expression-labelled tissues where construct was expressed, while ECFP expression-labelled tissues where CRISPR systems were active. The same microscope settings were maintained in between imaging different samples injected with DNA constructs. Embryos injected with iSBH-sgRNAs without triggers were genotyped following injection to confirm the presence of dCas9-Vp64 and the 8xCTS-ECFP reporters. Due to the strength of the signal, laser power had to be decreased for samples injected with chemically modified sgRNAs.

## Acknowledgements

We would like to acknowledge Dr Martyna Lucoseviciute and Dr Filipa Simoes for performing embryonic microinjections necessary for the generation of zebrafish transgenic lines and for mentorship with the zebrafish experiments. We would also like to thank Dr Vanessa Chong-Morrison and Prof Martin Jinek for providing suggestions that improved this manuscript. Dr Vanessa Chong-Morrison also helped with designing zebrafish expression plasmids and provided guidance on how to perform embryonic microinjections. Dr David Knapp also provided advice for this project and was a tremendous support in performing RNA circularisation assays. We would also like to acknowledge Jennifer Stott and Sasha Thomas from Integrated DNA Technologies for technical advice on chemically modified sgRNAs. We would also like to thank all present and past members of the TAF and TSS labs. OP was funded by the EPSRC & BBSRC Centre for Doctoral Training in Synthetic Biology, University of Oxford (grant EP/L016494/1), EvOX Therapeutics, and Wadham College. TSS, SM, and OP were funded by the Wellcome Trust (215615/Z/19/Z), and TSS also received institutional support from the Stowers Institute for Medical Research.

# Additional information

## Funding

| Funder | Grant reference number | Author |
|---|---|---|
| UKERC | grant EP/L016494/1 | Oana Pelea |
| Wellcome Trust | 10.35802/215615 | Oana Pelea<br>Tatjana Sauka-Spengler |
| Stowers Institute for Medical Research | | Tatjana Sauka-Spengler |

The funders had no role in study design, data collection and interpretation, or the decision to submit the work for publication. For the purpose of Open Access, the authors have applied a CC BY public copyright license to any Author Accepted Manuscript version arising from this submission.

## Author contributions

Oana Pelea, Conceptualization, Software, Formal analysis, Validation, Investigation, Visualization, Methodology, Writing – original draft; Sarah Mayes, Investigation; Quentin RV Ferry, Conceptualization, Supervision, Writing – review and editing; Tudor A Fulga, Conceptualization, Supervision; Tatjana Sauka-Spengler, Conceptualization, Supervision, Funding acquisition, Methodology, Writing – review and editing

## Author ORCIDs

Oana Pelea ⬥ http://orcid.org/0000-0002-1968-9827
Sarah Mayes ⬥ http://orcid.org/0000-0001-7630-9272
Quentin RV Ferry ⬥ http://orcid.org/0000-0003-1843-8572
Tudor A Fulga ⬥ http://orcid.org/0000-0002-1056-0082
Tatjana Sauka-Spengler ⬥ https://orcid.org/0000-0001-9289-0263

## Ethics

Zebrafish experiments were carried out according to regulated procedures authorised by the UK Home Office within the framework of the Animals (Scientific Procedures) Act 1986. Embryos used were derived from AB zebrafish strains.

Reviewer #1 (Public Review): https://doi.org/10.7554/eLife.87722.3.sa1
Author response https://doi.org/10.7554/eLife.87722.3.sa2

# Additional files

## Supplementary files

MDAR checklist

Supplementary file 1. Supplementary Materials – relevant RNA and primer sequences.

## Data availability

All DNA and RNA sequences used in this study are provided in the *Supplementary file 1*, while plasmids necessary for replicating this study have been deposited to AddGene: https://www.addgene.org/browse/article/28234145/. Computational pipelines for iSBH-sgRNA designs https://github.com/OanaPelea/Design_tools_iSBH-sgRNAs (*Pelea, 2025b*) as well as Flow Cytometry data analysis https://github.com/OanaPelea/Flow_Cytometry_Data_Analysis (*Pelea, 2025a*) are hosted on GitHub.

The following dataset was generated:

| Author(s) | Year | Dataset title | Dataset URL | Database and Identifier |
|---|---|---|---|---|
| Pelea O, Mayes S, Ferry QRV, Fulga TA, Sauka-Spengler T | 2023 | Specific Modulation of CRISPR Transcriptional Activators through RNA-Sensing Guide RNAs in Mammalian Cells and Zebrafish Embryos | https://www.addgene.org/browse/article/28234145/ | Addgene, 28234145 |

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
