## [Editor Report · eLife assessment]

The authors aim to develop a CRISPR system that can be activated upon sensing an RNA. As an initial step to this goal, they describe RNA-sensing guide RNAs for controlled activation of CRISPR modification. Many of the data look **convincing** and while several steps remain to achieve the stated goal in an in vivo setting and for robust activation by endogenous RNAs, the current work will be **important** for many in the field.

---

## [Referee Report · Reviewer #1 (Public Review)]

This paper describes RNA-sensing guide RNAs for controlled activation of CRISPR modification. This works by having an extended guide RNA with a sequence that folds back onto the targeting sequence such that the guide RNA cannot hybridise to its genomic target. The CRISPR is "activated" by the introduction of another RNA, referred to as a trigger, that competes with this "back folding" to make the guide RNA available for genome targeting. The authors first confirm the efficacy of the approach using several RNA triggers and a GFP reporter that is activated by dCas9 fused to transcriptional activators. A major potential application of this technique is the activation of CRISPR in response to endogenous biomarkers. As these will typically be longer than the first generation triggers employed by the authors they test some extended triggers, which also work though not always to the same extent. They then introduce MODesign which may enable the design of bespoke or improved triggers. After that, they determine that the mode of activation by the RNA trigger involves cleavage of the RNA complexes. Finally, they test the potential for their system to work in a developmental setting - specifically zebrafish embryos. There is some encouraging evidence, though the effects appear more subtle than those originally obtained in cell culture.

Overall, the potential of a CRISPR system that can be activated upon sensing an RNA is high and there are a myriad of opportunities and applications for it. This paper represents a reasonable starting point having developed such a system in principle.

The weakness of the study is that it does not demonstrate that the system can be used in a completely natural setting. This would require an endogenous transcript as the RNA trigger with a clear readout. The authors now acknowledge this limitation in their revised manuscript. Future studies and experiments should focus on these aspects in order for the system to be employed to its full and intended potential.

---

## [Author Response]

The following is the authors’ response to the original reviews.

Below, we provide a detailed account of the changes we made. For clarity and ease of review:

• Original reviewers' comments are included and highlighted in grey

• Our responses to each comment are written in black text

• Print screens illustrating the specific changes made to the manuscript are enclosed within black squares

**eLife assessment**
The authors aim to develop a CRISPR system that can be activated upon sensing an RNA. As an initial step to this goal, they describe RNA-sensing guide RNAs for controlled activation of CRISPR modification. Many of the data look convincing and while several steps remain to achieve the stated goal in an in vivo setting and for robust activation by endogenous RNAs, the current work will be important for many in the field.

The eLife assessment summarises our ambition to create a CRISPR system controlled by RNA sensing. The synopsis provided encapsulates the essence of our research, emphasising both the progress we have made and the challenges that lie ahead. This assessment fully resonates with our views.

**Public Reviews:**

**Reviewer #1 (Public Review):**
This paper describes RNA-sensing guide RNAs for controlled activation of CRISPR modification. This works by having an extended guide RNA with a sequence that folds back onto the targeting sequence such that the guide RNA cannot hybridise to its genomic target. The CRISPR is "activated" by the introduction of another RNA, referred to as a trigger, that competes with this "back folding" to make the guide RNA available for genome targeting. The authors first confirm the efficacy of the approach using several RNA triggers and a GFP reporter that is activated by dCas9 fused to transcriptional activators. A major potential application of this technique is the activation of CRISPR in response to endogenous biomarkers. As these will typically be longer than the first generation triggers employed by the authors they test some extended triggers, which also work though not always to the same extent. They then introduce MODesign which may enable the design of bespoke or improved triggers. After that, they determine that the mode of activation by the RNA trigger involves cleavage of the RNA complexes. Finally, they test the potential for their system to work in a developmental setting - specifically zebrafish embryos. There is some encouraging evidence, though the effects appear more subtle than those originally obtained in cell culture.Overall, the potential of a CRISPR system that can be activated upon sensing an RNA is high and there are a myriad of opportunities and applications for it. This paper represents a reasonable starting point having developed such a system in principle.The weakness of the study is that it does not demonstrate that the system can be used in a completely natural setting. This would require an endogenous transcript as the RNA trigger with a clear readout. Such an experiment would clearly strengthen the paper and provide strong confidence that the method could be employed for one of the major applications discussed by the authors. The zebrafish data relied on exogenous RNA triggers whereas the major applications (as I understood them) would use endogenous triggers.Related, most endogenous RNAs are longer than the various triggers tested and may require extensive modification of the system to be detected or utilised effectively.While additional data would clearly be beneficial, there should nevertheless be a more detailed discussion of these caveats and/or the strengths and applications of the system as it is presented (i.e. utility with synthetic triggers).

We agree with the observation regarding the subtler effects in the zebrafish embryos and the reliance on exogenous RNA triggers. Indeed, the utilisation of endogenous transcripts as triggers in a natural setting is a logical next step. We further acknowledge the need to delve deeper into the complexities and challenges of our system, particularly concerning the detection of endogenous RNA, thus offering valuable insights for researchers looking to adapt our system for various applications. In order to clarify these limitations, we made some changes in the final version of our paper. The following paragraphs have been therefore included in the manuscript discussion:

“In their current iteration, iSBH-sgRNAs show considerable promise for mammalian synthetic biology applications. Specifically, their ability to detect synthetic triggers could be pivotal in the development of complex synthetic RNA circuits and logic gates, thereby advancing the field of cellular reprogramming. However, further work is required to achieve better ON/OFF activation ratios in vivo and more homogeneous activity across tissues in the presence of RNA triggers. Additional chemical modifications could improve iSBH-sgRNA properties, and we believe that chemical modification strategies adopted for siRNA drugs or antisense oligos (Khvorova and Watts (2017)) could also be essential for further iSBH-sgRNA technology development. As iSBH-sgRNAs might be targeted by endogenous nucleases, leading to their degradation, a strategy for preventing this could involve additional chemical modifications. When inserted at certain key positions, such modifications could prevent interaction between iSBH-sgRNAs and cellular enzymes by introducing steric clashes or inhibiting RNA hydrolysis.

Once achieving superior dynamic ranges of iSBH-sgRNA activation in vivo, the next steps would involve understanding the classes of endogenous RNAs that could act as triggers. The chances that an iSBH-sgRNA encounters an endogenous RNA trigger inside a cell would depend on the relative concentrations of the two RNA species. Therefore, a first step towards determining potential endogenous RNA triggers will involve identifying RNA species with comparable expression levels as iSBH-sgRNAs. Then, iSBH-sgRNAs could be designed against these RNA species, followed by experimental validation. It is important to note that eukaryotic cells express a wide range of transcripts of varying sizes, expression levels, and subcellular localisations, all of which could greatly affect iSBH-sgRNA activation levels. Based on the data presented here, we speculate that RNA species up to 300nt that are also highly expressed might act as good triggers. Furthermore, as sgRNAs are involved in targeting Cas9 to genomic DNA in the nucleus, attempting to detect transcripts that are sequestered in the nucleus might also provide additional benefit.”

**Reviewer #3 (Public Review):**
In this work, the authors describe engineering of sgRNAs that render Cas9 DNA binding controllable by a second RNA trigger. The authors introduce several iterations of their engineered sgRNAs, as well as a computational pipeline to identify designs for user-specified RNA triggers which offers a helpful alternative to purely rational design. Also included is an investigation of the fate of the engineered sgRNAs when introduced into cells, and the use of this information to inform installation of modified nucleotides to improve engineered sgRNA stability. Engineered sgRNAs are demonstrated to be activated by trigger RNAs in both cultured mammalian cells and zebrafish.The conclusions made by the authors in this work are predominantly supported by the data provided. However, some claims are not consistent with the data shown and some of the figures would benefit from revision or further clarification.Strengths:- The sgRNA engineering in this paper is performed and presented in a systematic and logical fashion.- Inclusion of a computational method to predict iSBH-sgRNAs adds to the strength of the engineering.- Investigation into the cellular fate of the engineered sgRNAs and the use of this information to guide inclusion of chemically modified nucleotides is also a strength.- Demonstration of activity in both cultured mammalian cells and in zebrafish embryos increases the impact and utility of the technology reported in this work.Weaknesses:- While the methods here represent an important step forward in advancing the technology, they still fall short of the dynamic range and selectivity likely required for robust activation by endogenous RNA.- While the iSBH-sgRNAs where the RNA trigger overlaps with the spacer appear to function robustly, the modular iSBH-sgRNAs seem to perform quite a bit less well. The authors state that modular iSBHsgRNAs show better activity without increasing background when the SAM system is added, but this is not supported by the data shown in Figure 3D, where in 3 out of 4 cases CRISPR activation in the absence of the RNA trigger is substantially increased.- There is very little discussion of how the performance of the technology reported in this work compares to previous iterations of RNA-triggered CRISPR systems, of which there are many examples.

Concerning the methods falling short of the dynamic range and selectivity required for robust activation by endogenous RNA, we acknowledge this limitation and recognise the need for improvement in this area. In the resubmitted version of the manuscript, we provided a detailed discussion on how the selection of appropriate triggers might partially improve dynamic ranges and selectivity. This includes an exploration of various strategies and considerations that may enhance the robustness of our system (print screen above, also used for addressing Reviewer #1 comments).

Regarding the inconsistent performance of the modular iSBH-sgRNAs, we acknowledge that modular iSBH-sgRNAs seem to perform slightly less well than first- and second-generation designs. In order to illustrate this, we modified corresponding bar graphs to include fold turn-on iSBH-sgRNA activation in addition to significance (Figures 1, 2 and 3 of the manuscript). We also acknowledge this fact in the text, as well as we recognise this discrepancy in the Figure 3.D and provide further clarifications. To help conveying this message even further, we introduced a new figure (Figure 3- figure supplement 2) to accompany the heat map shown in the Figure 3.D. with corresponding bar graphs. These changes are documented below:

“…promoters. We ran 11 MODesign simulations for each trigger, incrementally extending the loop size while keeping the sgRNA 2 spacer input constant. HEK293T validation experiments showed that choosing modular iSBH-sgRNAs that detect the 4 U6-expressed triggers is possible (Figure 3.D, Figure 3- figure supplement 1.C). Despite not performing quite as well as second-generation designs (Figure 2.A.,Figure 3.D),modular iSBH-sgRNA still enable efficient RNA detection, especially for smaller RNAs such as triggers A and D. For highly efficient designs such asmodular iSBH-sgRNA (D), addition of the SAM effector system (Konermann et al. (2015)) boosted ON-state activation with only a negligible increase in the the OFF-state non-specific activation. Orthogonality tests suggested that activation of modular iSBH-sgRNA designs was specifically conditioned by complementary RNA triggers (Figure 3.E, Figure 3 - figure supplement 2), showing the exquisite specificity of the system.”

**Author response image 1. sa2fig1:** This supplementary figure reinterprets the data presented in Figure 3. E. using bar plots for enhanced clarity and comparison. It depicts the results of cotransfecting HEK293T cells with four modular iSBH-sgRNAs (A, B, C, and D) and examines all combinations of iSBH-sgRNA: RNA trigger pairings. The bar plots provide a visual representation of mean values with error bars indicating the standard deviation, based on three biological replicates.

Regarding the concern about the lack of comparison with previous iterations of RNA-triggered CRISPR systems, we also acknowledged other similar technologies within the discussion. We also point readers to a literature review we recently published (doi/full/10.1089/crispr.2022.0052) where we describe other similar technologies in more detail.

“To date, a variety of RNA-inducible gRNA designs have been developed (Hanewich-Hollatz et al. (2019); Hochrein et al. (2021); Jakimo et al. (2018); Jiao et al. (2021); Jin et al. (2019); Li et al. (2019); Liu et al. (2022); Lin et al. (2020); Siu and Chen (2019); Galizi et al. (2020); Hunt and Chen (2022b,a); Ying et al. (2020); Choi et al. (2023)). Nevertheless, there is a lack of direct, head-to-head comparisons of these designs under standardised experimental conditions. Some designs were evaluated in vitro, others in bacterial systems, and some in mammalian cells. Consequently, it is challenging to conclusively determine which design exhibits superior properties (Pelea et al. (2022)). Notably, to the best of our knowledge, the iSBH-sgRNA systemis the first RNA-inducible gRNA design tested in vivo and characterising the iSBH-sgRNA activation mechanism was essential for implementing iSBH-sgRNA technology in zebrafish embryos. In vivo, chemical modifications in the spacer sequence were vital for iSBH-sgRNA stability and function.”